# Development, Optimization, and Comparison of Different Sample Pre-Treatments for Simultaneous Determination of Vitamin E and Vitamin K in Vegetables

**DOI:** 10.3390/molecules25112509

**Published:** 2020-05-28

**Authors:** Antonella Aresta, Gualtiero Milani, Maria Lisa Clodoveo, Carlo Franchini, Pietro Cotugno, Ivana Radojcic Redovnikovic, Maurizio Quinto, Filomena Corbo, Carlo Zambonin

**Affiliations:** 1Department of Chemistry, University “Aldo Moro”, Bari Via E. Orabona, 4, I-70125 Bari, Italy; antonellamaria.aresta@uniba.it (A.A.); pietro.cotugno@uniba.it (P.C.); carlo.zambonin@uniba.it (C.Z.); 2Department of Pharmacy-Pharmaceutical Sciences, University “Aldo Moro”, Bari Via E. Orabona, 4, I-70125 Bari, Italy; gualtiero.milani@uniba.it (G.M.); carlo.franchini@uniba.it (C.F.); 3Interdisciplinary Department of Medicine, University “Aldo Moro”, Bari Pz G. Cesare 11, Policlinico di Bari, 70124 Bari, Italy; marialisa.clodoveo@uniba.it; 4Faculty of Food Technology and Biotechnology, University of Zagreb, Pierottijev 6, 10 000 Zagreb, Croatia; irredovnikovic@pbf.hr; 5Department of Agricultural Food and Environmental Sciences, University of Foggia, Via A.Gramsci 89/91, 71122 Foggia, Italy; maurizio.quinto@unifg.it

**Keywords:** SPME, GC–MS, α-tocopherol, α-tocopheryl acetate, phylloquinone, menaquinone-4, vegetables

## Abstract

The absence of vitamin E from the diet can lead to cardiovascular disease, cancer, cataracts, and premature aging. Vitamin K deficiency can lead to bleeding disorders. These fat-soluble vitamins are important nutritional factors that can be determined in different methods in vegetables. In this work, the simultaneous determination of α-tocopherol, α-tocopheryl acetate, phylloquinone, and menaquinone-4 by gas chromatography–mass spectrometry (GC–MS) has been optimized using both direct injection and solid phase microextraction (SPME). Three different sample pre-treatment approaches based on: (A) solid–liquid–liquid–liquid extraction (SLE–LLE), (B) SLE, and (C) SPME were then applied to extract the target analytes from vegetables samples using menaquinone as internal standard. All the procedures allowed the determination of the target analytes in onion, carrot, celery, and curly kale samples. Similar results were obtained with the three different approaches, even if the one based on SPME offers the best performance, together with a reduced use of solvent, time consumption, and experimental complexity, which makes it the preferable option for industrial applications.

## 1. Introduction

The determination of natural products of interest to human health (bioactive compounds) is carried out through a complex workflow that generally includes many steps such as extraction, fractionation, separation, and isolation of the compounds. However, the complex nature of natural extracts and low levels are often an obstacle to their correct determination.

The traditional strategies to isolate bioactive compounds from vegetables are based on liquid–liquid extraction (LLE), solid–liquid extraction (SLE), or solid phase extraction (SPE) using solvents of different polarity. The first two are very useful techniques for isolating, pre-concentrating and transferring analytes in a phase that is compatible with the analytical instrument. These extraction techniques are based on passive diffusion principle through the interface between the two phases, the extraction times are linked to interfacial contact, generally a greater agitation determines a reducing of times. The increase in stirring speed simultaneously increases the risk of forming emulsions which will in turn require centrifugation, filtration or addition of other solvent or substances to disrupt the emulsion. SPE requires a lower use of organic solvent and is based on systems such as: cartridges, columns, or membranes containing an extractive absorbent. The sample is inserted into the cartridge and all the matrix interferences that have been adsorbed are washed by a weak solvent and the elution of the analytes is done using a small amount of the appropriate solvent [1]. The crucial factors to apply these analytical techniques are the choice of the solvent linked to the affinity for analytes, the toxicity, the selectivity, the volatility, and finally the price, which must not be underestimated.

The search for fast, efficient, cost effective, and environment friendly methods of analysis prompted the introduction in 1989 of Solid Phase Microextraction (SPME) by J. Pawliszyn [2]. This technique compared to conventional ones is faster, simpler, sensitive and is fully eco-sustainable by requiring only water for the extraction. Since the invention of the technique until today, its applications have dramatically increased. Therefore, to date, its applications are numerous, and include the determination of natural products present in trace in drinking, fruits and organic samples [2]. Finally, dispersive solid phase extraction (d-SPE) analysis is a simple and straightforward sample preparation technique suitable for a wide variety of food and agricultural products. One specific method which has become popular is the QuEChERS (Quick, Easy, Cheap, Effective, Rugged e Safe) technique which was created to facilitate the rapid screening of large numbers of food and agricultural samples for pesticide residues.

It could be a valid alternative to SPME, but according to our knowledge it is not among the methods of choice for the determination of fat-soluble vitamins.

In the light of these considerations, in this paper, the SPME has been chosen for the scope of the experimental work described in this paper that aims to develop a rapid, sensitive, and specific analytical method for the simultaneous determination of fat-soluble vitamins E and K in food and biological matrices.

Fat-soluble vitamins E and K are important nutritional factors and their determination can be useful for nutraceutical, nutrigenomics, and nutribolomics studies. The absence of vitamin E from the diet can lead to cardiovascular disease, cancer, cataracts, and premature aging [3,4]. Vitamin K deficiency can lead to bleeding disorders [5]. Humans and animals do not synthesize Vitamin E, thus, α- tocopherol, according to Directive N. 95/2/EC (1995) of the European Parliament, is an official food additive or dietary supplement. The importance of Vitamin E is also documented by a specific Health Claim [6,7,8] approved by the EFSA (European Food Safety Authority) that can be used to label functional foods with a high healthy value, e.g., extra virgin olive oil rich with Vitamin E.

Tocopheryl acetate (vitamin E acetate) is the ester of acetic acid and tocopherol is mainly used in dermatological products. Vitamin K includes different compounds that are characterized by a naphthoquinone ring and a different aliphatic side chain that define Vitamin K1 (phylloquinone) and Vitamin K2 (menaquinones). Phylloquinone is made by plants, mostly green leafy vegetables, and is widely used as supplement to treat certain bleeding disorders [5,9,10]. To date, the determination of α- tocopherol, tocopheryl acetate, phylloquinone, and menaquinones has been only accomplished separately. Historically, vitamin E was considered difficult to measure because of its instability and its lipophilicity, but these challenges have largely been overcome with modern technologies [11].

Existing methods for the determination of vitamins in food are mainly based on traditional extraction techniques, mainly solid phase extraction (SPE), followed by chromatographic analysis [12,13,14]. Pressurized liquid extraction (PLE) and dispersive liquid–liquid microextraction (DLLME) were also used to isolate and preconcentrate tocopherols and tocotrienols from plant foods [15]. A good, simple, and inexpensive alternative has been proposed by Aresta et al. based SPME coupled to GC–MS for the determination of α-tocopherol in olive oil as other important bioactive compound in different food matrix [16,17,18,19].

A fast and simple method for the determination of vitamins, including vitamin E acetate, in infant formula by DLLME combined with HPLC–UV has been also recently reported [20]. Very recently, an overview of methods for assessment of vitamin K has been provided by Górska [21]. Existing methods are usually laborious and there is still a great need for the development of robust, reliable methods for the quantitative determination of vitamin K. For instance, SPE–LC approaches were proposed for the determination of Vitamin K1 in fish feeds and shrimp post larvae [22] and edible microgreens [23] while DLLME coupled to liquid chromatography with diode-array and ESI (Electrospray ionization)-mass spectrometry detection (proved to be useful for routine control analysis of vitamins D and K in different food commodities [24]. Lately, two approaches based on liquid and solid phase extraction followed by ultra-high-performance liquid chromatography–atmospheric pressure chemical ionization tandem mass spectrometric (UHPLC–APCI-MS/MS) and liquid chromatography–with diode array-fluorescence detector (HPLC–DAD–FLD) were successfully developed for the determination of eight vitamin K compounds in 17 fermented food products at micrograms level [25] and of tocols, γ-oryzanols, phytosterols, squalene, cholecalciferol, and phylloquinone in rice bran and vegetable oil samples [26], respectively. A faster method for the determination of vitamin K in leaves was based on SPME–GC–FID using a polydimethylsiloxane (PDMS) fiber [27].

In the light of these considerations in the present work, the task of the simultaneous determination of α-tocopherol, α-tocopheryl acetate, phylloquinone and menaquinone by gas chromatography–mass spectrometry has been addressed. Three different sample pre-treatment approaches, of which the first two based on conventional extraction techniques and the third on SPME, were applied to extract the target analytes from vegetables samples (onion, carrot, celery, and curly kale) using menaquinone as the internal standard. All of the procedures permitted the determination of the target analytes in the selected food at satisfactory LODs, limits of detection, even if SPME proved to be the simpler and faster approach.

## 2. Results

### 2.1. Vegetable Samples

All samples (onion, carrot, celery, and curly kale) were of local origin and collected from supermarkets. Each vegetable was finely grated and immediately subject to extraction procedures or dried for preservation. In this regard, the grated sample was portioned into aliquots (0.5 g) that were dried in a CHRIST-vacuum concentrator (Centrifuge Concentrator Centrifuge) (RVC 2–18, Osterode am Harz, Germany) at 30 °C until reduced to 43% ± 1% their weight. Dry residues were stored at 8 °C in the dark until analyzed. Spiked samples were prepared in triplicate by adding suitable amounts of standard solutions to fresh or dry aliquots.

### 2.2. Programmed Temperature Vaporization (PTV) and GC Parameter Optimization Using Direct Injection

The method was initially developed working in direct injection (1 µL) mode. Initially, the parameters of the PTV injector were optimized. The low starting temperature (70 °C) avoided the overflow of the liner with solvent vapors, the heating rate (14.5 °C/s) and the splitless time of 1 min optimized the transfer of the analytes into the column, the final temperature (300 °C) assured the cleaning of the inlet. The chromatographic conditions were then optimized to achieve good separation and chromatographic efficiency. The GC–MS chromatogram was obtained by directly injecting a standard of α-tocopherol, α-tocopheryl acetate, phylloquinone, and menaquinone-4 at a concentration level of 50 µg mL^−1^ using the optimized temperature gradient program and shows its resolution capability as reported in Figure 1. Calibration curves were linear in the explored concentration range with correlation coefficients greater than 0.999 and intercepts significantly close to zero at the 95% confidence level. The within-day (*n* = 5) and between-days (*n* = 5 over 5 days) coefficients of variation, estimated by an ANOVA test, ranged from 2.8 (α-tocopherol) to 4.0 (menaquinone) % and from 3.4 (α-tocopherol) to 5.9 (menaquinone) %, respectively, and remained practically unchanged at 5 and 50 µg mL^−1^. The limit of detection (LOD) and limit of quantification (LOQ) were calculated as three and ten-fold the standard deviation of the intercept of the calibration curves. The estimated LODs were in the ranges 0.1 (α-tocopherol and α-tocopherol acetate)–0.6 µg mL^−1^ (menaquinone-4), respectively, while LOQs were in the range 0.4 (α-tocopherol)–1.9 µg mL^−1^ (menaquinone-4), respectively. All the validation parameters are resumed in Table 1. The within day and between days RSD% were always better than 4.0 and 5.9, respectively. 

### 2.3. SPME Procedure Optimization

Experiments were then devoted to the optimization of the SPME conditions and to the validation of the method using SPME. A 7-μm thick PDMS-coated fiber was chosen for the simultaneous extraction of the selected vitamins from aqueous solutions since it has been successfully employed [17,26,27] for the extraction of α-tocopherol, vitamin K1 and KIt was also reported that the moderate addition of ethanol in water increases the solubility of the analytes [28] due to lower hydrophobic repulsion between ethanol and vitamin molecules, while the addition of HCl (10 mM) to the water–ethanol mixture [17,29] has a positive effect on the extraction of α-tocopherol with the PDMS fiber. Thus, the effect of the addition of ethanol and HCl, respectively, on the extraction efficiency were evaluated. Figure 2 reports how the extraction efficiency was affected by the addition of varying concentrations of ethanol and HCl (10 mM). While the addition of 10% ethanol has a positive effect on the extraction of all the analytes, a further increase in its concentration (20%) produced opposite results, i.e., response increase for the more hydrophobic vitamins E and response decrease for vitamins K. The addition of HCl shows rather positive results for all the investigated compounds. Consequently, a concentration of 10% ethanol with the presence of HCl (10 mM) was used for further experiments. Usually, salt addition increases the ionic strength of the solution and the organic compounds become less soluble increasing the partition coefficients between the phases. Four different concentrations of NaCl (0, 10%, 20%, and 30%) in water with 10% ethanol and 10 mM HCl were tested and the relevant results in terms of extraction efficiency are shown in Figure 2. It was apparent, in agreement with Aresta et al. [17,29] but in contrast with [27] that the extraction efficiency for all the analytes was negatively affected by salt addition and was not further investigated.

As far as the extraction volume is concerned, 15 mL was selected as optimal value, since it was found that lower volumes produced a response decrease for all the analytes, while no effects were observed using higher volumes. Figure 3 shows the extraction time profiles obtained on Vitamins E and K at the temperature of 20 and 50 °C, respectively. The equilibrium conditions were almost always reached after 30 min (50 °C) and 40 min (20 °C) of extraction and the higher extraction efficiencies were obtained at 50°C for all the analytes except for α-tocopherol that showed a significant response decrease and, according to the literature [17,29] never reached equilibrium conditions. The extraction time of 30 min and the temperature of 50 °C were selected for the following experiments. The desorption temperature should be high enough to ensure the complete desorption of the extracted compounds from the fiber coating. In the present work, desorption was carried out for 5 min at 300 °C, since a significant “carry-over” was observed when desorbing at lower temperatures or shorter desorption times. The method was validated using the optimized SPME conditions and the relevant results are reported in Table 2. The dynamic range of the developed SPME-GC–MS procedure resulted in linear from the LOQ values for over two orders of magnitude, with correlation coefficients better than 0.997 and intercepts not significantly different from zero at 95% confidence level. The within day and between days RSD% were always better than 4.8 and 8.8, respectively. The estimated LOD and LOQ, calculated as three and ten-fold the standard deviation of the intercept of the calibration curves, were always better than 10 and 40 ng mL^−1^, respectively.

## 3. Discussion

The three sample pre-treatment approaches were applied on carrot, celery, curly kale, and onion for the simultaneous determination of α-tocopherol, α-tocopheryl acetate, and phylloquinone, using menaquinone-4 as the internal standard, since it is known to be absent in these vegetables. Figure 4 reports, as an example, the GC–MS extracted-ion chromatograms of a carrot dry sample pre-treated with the three procedures (A, B, and C) optimized in this work. The insets show the spectra acquired in the samples compared to those of the NIST (National Institute of Standards and Technology) library. As apparent, all the described extraction procedures permitted the easy determination of all the analytes and the internal standard. Quantitation was performed with the standard addition method. Recovery studies were performed at the concentration levels of 5, 50, and 100 µg mL^−1^ and the relevant results are shown in Table 3.

SPME (C) was not considered in these experiments since it is notoriously a non-exhaustive technique. Recoveries were found to be concentration independent (*p* < 0.05) and procedure B provides the highest recoveries (>80%). The matrix effect for the SPME procedure [30] was calculated as described in the experimental section, and the values were 69 ± 15, 80 ± 12 and 96 ± 5 for each dilution ratio (1:1 or 0.1:1 or 0.005:1, *w*/*v*, respectively). As apparent, the matrix effect was minimized working at the dilution ratio of 0.005:1. The optimized conditions are reported in the experimental section. The concentrations of the target vitamins were estimated in several fresh or dry samples; Table 4 reports, as an example, the results obtained in specific carrot, celery, curly kale and onion samples while Table 5 resumes the whole results reporting the concentration ranges found in all the samples. Finally, the accuracy of the SPME–GC–MS method was evaluated. Vitamins were added to each sample at twice the estimated concentrations or LOQs levels. The concentrations obtained were compared with expectations. The results were between 10 (menaquinone, carrot) and 3% (phylloquinone, onion) with an average of 6.5% ± 2.6%. Accuracy corresponded to the acceptance criteria, because the deviations of the mean values from target values were always below 15%.

## 4. Materials and Methods

### 4.1. Chemicals

All chemicals used were supplied by Sigma-Aldrich (Milan, Italy) and were analytical grade. The standard of vitamins E (dl-α-tocopherol), E acetate (dl-α-tocopheryl acetate), K1 (phylloquinone) and K2 (menaquinone-4) have a purity level greater than 96%. Vitamin stock solutions (0.4 mg mL^−1^) were made in 50% (*v*/*v*) acetonitrile in ethanol and stored in aliquots (0.5 mL) at −20 °C in the dark. More dilute working solutions were obtained just before use in the range 0.05–150 µg mL–SPME working solutions were prepared in water with 10% ethanol, HCl (10 mM) just before use.

### 4.2. Sample Pre-Treatment

Vitamins were extracted from vegetables using three different approaches. All samples were always added with the internal standard before the sample pretreatment. In the first approach (A), each sample (1 g of fresh or 0.1 g of dry sample) was homogenized (Ultra Turbax IKA T18 Basic) (Merck Life Science S.r.l. 20149 Milano Italy) with 10 mL of water and 0.4 mL of 200 μg mL^−1^ menaquinone (internal standard) at the speed of 15.000 rpm for 1 min, after which, 15 mL of 6% (*v*/*v*) 2-propanol in hexane was added. The mixture was vortexed for 1 min and centrifuged (ALC Multispeed Refrigerated Centrifuge PK 121R) (Universal Resource Trading LTD, United Kingdom) at 3500 rpm for 5 min at 21 °C. The organic phase was then recovered and dried under a gentle stream of the residue was dissolved in 0.4 mL of pure hexane and 1 µL was injected in the GC–MS system.

For the second extraction procedure (B), each sample (1 g of fresh or 0.1 g of dry sample) was subjected to a solid-liquid extraction (30 min at room temperature in the dark) with 10 mL of 50% (*v*/*v*) acetonitrile in ethanol with 0.1% (*w*/*v*) butylated hydroxy anisole and 0.4 mL of internal standard under magnetic stirring. Then, the mixture was centrifuged for 5 min at 3500 rpm; the supernatant was transferred into an amber tube containing 0.25 g of Na_2_SO_4_ anhydrous, briefly shaken and 1 µL injected.

The third procedure was based on SPME (C). Samples (0.075 g, fresh or dry) were placed in 15 mL amber vials, weighed and 15 mL of water with 10% (*v*/*v*) ethanol, HCl (10 mM) and 30 μL of internal standard were added. The solution was stirred for 15 min at 50 °C, and finally subjected to SPME for 30 min at 50 °C under magnetic stirring. Desorption of the analytes was performed into the GC injector kept at the constant temperature of 300 °C in splitless mode with 5 min of sampling time. The fiber was removed from the injector after 8 min, clean and ready for the next extraction. Quantifications were based on internal standard method. The matrix effect on the SPME was calculated [28] as follows. Different amounts (1.5 g, 0.15 g or 0.075 g) of vegetable samples were placed in 15 mL amber vials, spiked with the analytes at the concentration level of 100 µg g^−1^ and added with 15 mL of the extraction solution. Each sample was mixed for 15 min at 50 °C before being subjected to SPME-GC–MS analysis. Matrix effects were calculated by comparing the area responses obtained for each dilution ratio (1:1 or 0.1:1 or 0.005:1, *w*/*v*), with standard solutions at the corresponding concentrations as a reference

### 4.3. Apparatus

The gas chromatography–mass spectrometry system consisted of a Finnigan Trace GC Ultra gas chromatograph equipped with a split/splitless programmed temperature vaporization (PTV) injector coupled to an ion trap mass spectrometer (FinniganPolarisQ A TRACE™ TR-5 GC Column (Thermo Fisher Scientific Milan, Italy) (30 m, 0.25 µm i.d., 0.25 µm film thickness, Thermo Fisher Scientific, Milan, Italy) was used with helium as carrier gas (flow rate 1 mL min^−1^). The injector i.d. was 1 mm. The oven temperature program was 3 min at 180 °C, from 180 to 300 °C at 25 °C min^−1^, and 13 min at 300 °C. When working with direct injection, the injector worked in PTV splitless mode, to avoid thermal degradation and/or adsorption. The relevant parameters are reported in Table 6. For SPME, the injector temperature and the GC transfer line were both set at 300 °C. The mass spectrometer was operated in the electron impact positive ion mode with a source temperature of 250 °C. The electron energy was 70 eV and the filament current 250 µA. Mass spectra were acquired in the *m*/*z* range 50–Detection of the vitamins were made from extracted ion chromatograms (*m*/*z* 165, for vitamin E and E acetate, and 186 and 225 for K1 and K2, respectively) obtained in total ion current (TIC) mode.

## 5. Conclusions

In conclusion, although the benefits of SPME (high productivity and low solvent consumption) are well known in academic research, the technology is not yet used as widely as it should by regulatory agencies, perhaps due to the perception that quantitative analysis is not solid. as much as traditional techniques. In this study, however, the equivalence of the non-exhaustive SPME with the exhaustive extractions of the conventional SLE and LLE techniques is demonstrated. Sample preparation is much simpler (0.075 g of sample in 15 mL of water with 10% ethanol) and faster (extraction time with the fiber of only 30 min, fully automatic) compared to the other two conventional procedures. Gas chromatography–mass spectrometry (GC–MS) was employed for first time for the analysis of α-tocopherol, α-tocopheryl acetate and phylloquinone in vegetables samples (onion, carrot, celery, and curly kale) using menaquinone-4 as the internal standard. Samples were extracted using different pre-treatment methods, i.e., (A) solid–liquid–liquid–liquid extraction (SLE–LLE), (B) SLE, and (C) SPME. Even if all the approaches allowed the determination of the analytes in the selected samples, SPME procedure provided the best performances, together with a reduced use of solvent, time consumption, and experimental complexity. In summary, in this manuscript a good proof-of-concept report of the combination of different methods with a high-throughput and “green” sample preparation approach is reported. This approach could be extended to more target compounds and more Vitamins (Vitamin A for instance) in future work.

## Figures and Tables

**Figure 1 molecules-25-02509-f001:**
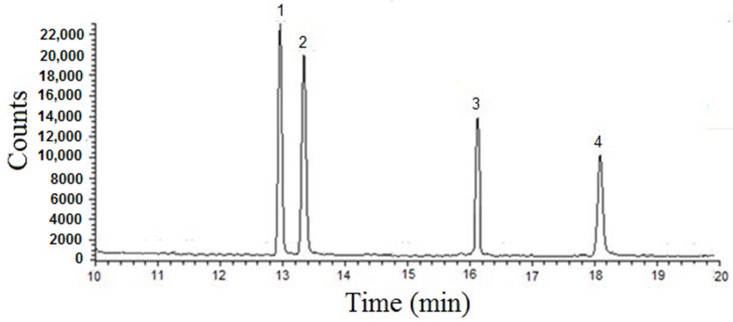
Gas chromatography–mass spectrometry (GC–MS) chromatogram obtained directly injecting a standard solution of α-tocopherol (**1**), α-tocopheryl acetate (**2**), phylloquinone (**3**), and menaquinone-4 (**4**) at a concentration level of 50 µg mL^−1^.

**Figure 2 molecules-25-02509-f002:**
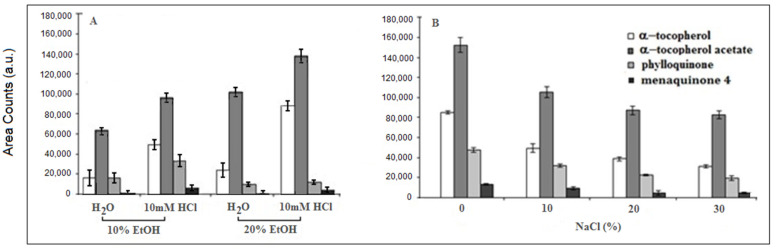
Effects of ethanol and HCl (**A**) and of NaCl (**B**) on the solid phase microextraction (SPME) efficiency.

**Figure 3 molecules-25-02509-f003:**
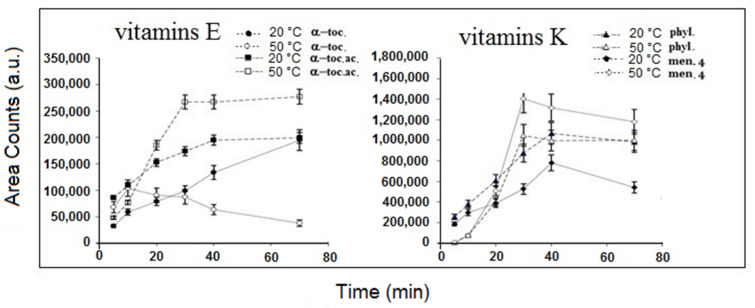
Extraction time profiles obtained on Vitamins E and K at the temperature of 20 and 50 °C, respectively.

**Figure 4 molecules-25-02509-f004:**
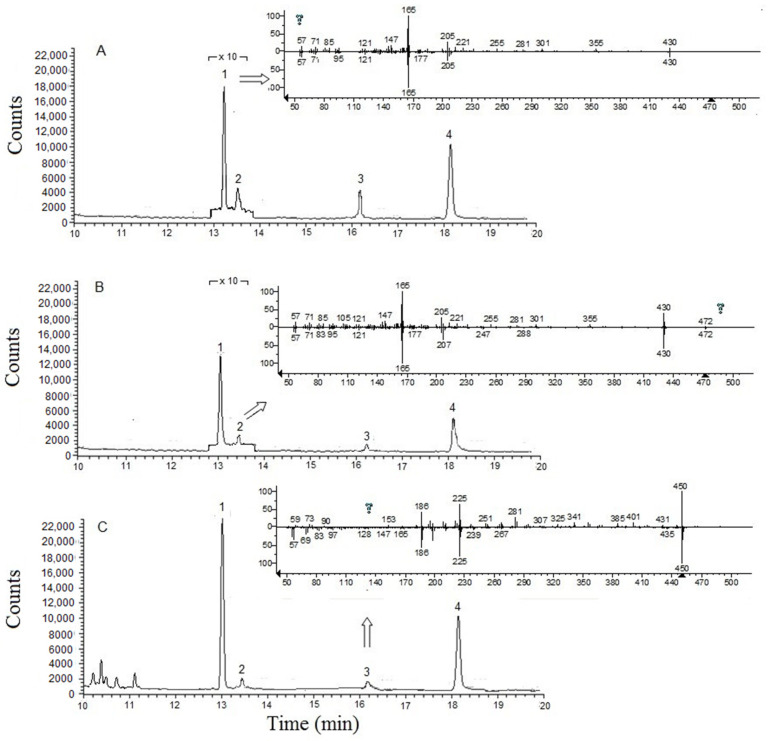
GC–MS extracted-ion chromatograms (*m*/*z* 165 and 186) of a carrot dry sample extracted by the pre-treatment (**A**), (**B**), and (**C**), (see Section 4.2) respectively. α-tocopherol (**1**), α-tocopheryl acetate (**2**), phylloquinone (**3**), and menaquinone-4 (**4**). To compare resolution in problematic segment in Figure 4A,B, a zoom (× 10) is plotted. The NIST library values for α-tocopherol, α-tocopheryl acetate and phylloquinone were 830, 850, and 813, respectively. The insets show the spectra acquired in the samples (**up**) compared to those of the NIST library (**down**).

**Table 1 molecules-25-02509-t001:** Linear range, detection limits and precision of α-tocopherol, α-tocopherol acetate, phylloquinone, and menaquinone-4 obtained by direct injection.

Analyte	Linearity Range (µg mL^−1^)	R^2^	LOD (µg mL^−1^)	LOQ (µg mL^−1^)	Within-Day (RSD%, *n* = 3)	Between-Days (RSD%, *n* = 15)
α-tocopherol	0.4–100	0.999	0.1	0.4	2.8	3.4
α-tocopherol acetate	0.4–100	0.999	0.1	0.4	2.9	3.6
Phylloquinone	0.7–100	0.999	0.2	0.7	3.9	4.4
Menaquinone	1.9–100	0.999	0.6	1.9	4.0	5.9

**Table 2 molecules-25-02509-t002:** Linear range, detection limits and precision of α-tocopherol, α-tocopherol acetate, phylloquinone, and menaquinone-4 obtained by SPME.

Analyte	Linearity Range (µg mL^−1^)	R^2^	LOD (µg mL^−1^)	LOQ (µg mL^−1^)	Within-Day (RSD%, *n* = 3)	Between-Days (RSD%, *n* = 15)
α-tocopherol	0.004–1.0	0.9999	0.001	0.004	4.3	8.8
α-tocopheryl acetate	0.006–1.0	0.9999	0.002	0.006	4.8	7.5
phylloquinone	0.036–5.0	0.9999	0.011	0.036	4.7	8.0
menaquinone	0.038–5.0	0.9990	0.011	0.038	4.6	6.9

**Table 3 molecules-25-02509-t003:** Percentage recoveries obtained with the sample pretreatments A and B.

Method	A			B		
[Vitamin]	[5] (µg mL^−1^)	[50] (µg mL^−1^)	[100] (µg mL^−1^)	[5] (µg mL^−1^)	[50] (µg mL^−1^)	[100] (µg mL^−1^)
α-tocopherol	59 ± 16%	61 ± 18%	63 ± 7%	84 ± 15%	93 ± 11%	85 ± 6%
α-tocopheryl acetate	55 ± 7%	58 ± 10%	60 ± 6%	83 ± 18%	77 ± 6%	80 ± 7%
phylloquinone	64 ± 15%	66 ± 6%	60 ± 12%	86 ± 12%	85 ± 13%	81 ± 9%
menaquinone	65 ± 24%	66 ± 10%	66 ± 23%	93 ± 21%	88 ± 18%	95 ± 10%

**Table 4 molecules-25-02509-t004:** Concentrations of the target vitamins estimated in specific carrot, celery, curly kale, and onion samples.

Vegetable		Analyte Concentration (µg g^−1^)					
		[α-tocopherol]		[α-tocopheryl acetate]		[phylloquinone]	
**carrot**		**fresh**	**dry**	**fresh**	**dry**	**Fresh**	**dry**
Pre-treatment	A	14.5 ± 1.4	32.0 ± 1.1	2.4 ± 0.4	6.0 ± 0.4	2.6	8.1 ± 0.1
	B	15.0 ± 2.5	30.6 ± 9.1	nd	5.9 ± 0.6	nd	7.8 ± 0.1
	C	19.3 ± 1.1	47.9 ± 3.8	nd	6.8 ± 0.5	nd	8.0 ± 0.1
**celery**		**fresh**	**dry**	**fresh**	**dry**	**Fresh**	**dry**
Pre-treatment	A	18.9 ± 0.9	32.8 ± 9.4	2.8 ± 0.4	4.7 ± 0.2	3.2 ± 0.2	8.0 ± 0.6
	B	22.5 ± 1.1	46.1 ± 8.2	2.0 ± 0.6	4.9 ± 0.3	3.4 ± 0.1	8.6 ± 0.5
	C	20.0 ± 0.9	50 ± 1.2	2.0 ± 0.1	4.0 ± 0.3	nd	8.8 ± 0.7
**curly kale**		**fresh**	**dry**	**fresh**	**dry**	**fresh**	**dry**
Pre-treatment	A	17.9 ± 1.0	39.2 ± 2.3	1.9 ± 0.6	3.3 ± 0.3	6.9 ± 0.6	13.9 ± 0.8
	B	18.9 ± 1.5	43.6 ± 7.6	2.0 ± 0.5	3.8 ± 1.3	8.1 ± 0.5	13.5 ± 0.6
	C	20.6 ± 1.2	35.2 ± 2.2	2.0 ± 0.5	4.1 ± 0.8	9.3 ± 0.8	14.6 ± 1.1
**onion**		**fresh**	**dry**	**fresh**	**dry**	**fresh**	**dry**
Pre-treatment	A	4.9 ± 1.1	10.4 ± 2.1	2.2 ± 0.6	4.1 ± 0.9	3.0 ± 0.9	7.9 ± 0.8
	B	4.1 ± 0.9	11.5 ± 2.0	2.0 ± 0.1	4.7 ± 0.5	3.4 ± 0.9	8.5 ± 0.9
	C	3.3 ± 0.4	10.9 ± 1.1	2.0 ± 0.9	4.3 ± 0.9	nd	9.0 ± 1.1

nd = not detectable.

**Table 5 molecules-25-02509-t005:** Concentration ranges found for the selected analytes in all the selected samples using the three optimized sample pre-treatment approaches.

Number Samples	Vegetable	Concentration Range (µg g^−1^)					
		[α-tocopherol]		[α-tocopheryl acetate]		[phylloquinone]	
**10**	**carrot**	**fresh**	**dry**	**fresh**	**dry**	**fresh**	**dry**
Pre-treatment	A	6.1–44.3	13.5–98.0	nd−20.2	nd−50.5	nd−2.6	nd−8.1
	B	6.3–47.5	13.0–97.5	nd−19.8	nd−49.6	nd−3.6	nd−9.0
	C	5.6–39.8	13.9–98.8	nd−19.9	nd−52.2	nd−3.2	nd−8.0
**4**	**celery**	**fresh**	**dry**	**fresh**	**dry**	**fresh**	**dry**
Pre-treatment	A	18.0–20.9	32.0 ± 44.4	nd−2.5	nd−4.9	nd−3.6	nd−9.1
	B	17.8–22.5	31.9–43.8	nd−2.7	nd−4.8	nd−3.6	nd−9.0
	C	18.1–21.8	32.1–45.0	nd−2.5	nd−5.0	nd−3.9	nd−9.8
**6**	**curly kale**	**fresh**	**dry**	**fresh**	**dry**	**fresh**	**dry**
Pre-treatment	A	8.6–20.5	18.0–43.1	nd−13.0	nd−22.1	nd−7.6	nd−15.7
	B	8.9–19.8	18.3–42.2	2.0 ± 0.5	nd−21.9	nd−7.5	nd−14.6
	C	8.2–20.6	19.5–43.6	2.0 ± 0.5	4.1 ± 0.8	nd−9.3	nd−16.0
**14**	**onion**	**fresh**	**dry**	**fresh**	**dry**	**fresh**	**dry**
Pre-treatment	A	2.3–6.0	5.1–14.6	nd−8.1	nd−19.9	nd−3.5	nd−8.7
	B	2.0–5.6	5.0–14.1	nd−8.0	nd−20.3	nd−3.6	nd−9.0
	C	2.1–6.3	4.9–14.9	nd−7.9	nd−19.7	nd−3.5	nd−9.5

nd = not detectable.

**Table 6 molecules-25-02509-t006:** Operative parameters of programmed temperature vaporization (PTV) splitless injection.

Injection Phase	Pressure (KPa)	Rate (°C/sec)	Temperature (°C)	Time (min)
injection	83		70	0.15
evaporation	83	6.7	100	0.3
transfer	210	14.5	270	1
cleaning		14.5	300	3

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
