# Peer review of "Development, Optimization, and Comparison of Different Sample Pre-Treatments for Simultaneous Determination of Vitamin E and Vitamin K in Vegetables"

_molecules, 2020, doi:10.3390/molecules25112509_

Round 1
Reviewer 1 Report
The manuscript by Aresta, et al presentd the analysis of fat-soluble vitamins from produce. This directly impacts the growing field of food safety and, as such, warrants publication. The contribution is well-written with good logical flow. The results support the conclusions. A few minor items:
- The research hypothesis, as alluded to in lines 45-46, is the development of a sample pretreatment method that is rapid, inexpensive, reproducible, etc. This should be elaborated upon in the Introduction. For example, what are the research goals in terms of LOD, reproducibility, time, etc.? What are the quantitative measures of the previously reported studied that make them deficient? In recent years, QuEChERS has been used for similar analysis and should be discussed.
2. Section 2.2 does not present results. It should be moved the the Materials and Methods section.
3. The chromatograms presented in Figure 4A and 4B show a baseline anomaly in the region around peaks 1 and 2. Any idea of the cause of this?
Author Response
1.Comment of reviewer: The manuscript by Aresta, et al presented the analysis of fat-soluble vitamins from produce. This directly impacts the growing field of food safety and, as such, warrants publication. The contribution is well-written with good logical flow. The results support the conclusions. A few minor items
The research hypothesis, as alluded to in lines 45-46, is the development of a sample pretreatment method that is rapid, inexpensive, reproducible, etc. This should be elaborated upon in the Introduction. For example, what are the research goals in terms of LOD, reproducibility, time, etc.? What are the quantitative measures of the previously reported studied that make them deficient? In recent years, QuEChERS has been used for similar analysis and should be discussed.
1.Answer : We thank the Reviewer for the constructive comments on our manuscript.
We improved the introduction with the following text:
The traditional strategies to isolate bioactive compounds from vegetables are based on liquid-liquid extraction (LLE), solid-liquid extraction (SLE) or solid phase extraction (SPE) using solvents of different polarity. The first two are very useful techniques for isolating, pre-concentrating and transferring analytes in a phase that is compatible with the analytical instrument. These extraction techniques are based on passive diffusion principle through the interface between the two phases, the extraction times are linked to interfacial contact, generally a greater agitation determines a reducing of times. The increase in stirring speed simultaneously increases the risk of forming emulsions which will in turn require centrifugation, filtration or addition of other solvent or substances to disrupt the emulsion. SPE requires a lower use of organic solvent and is based on systems such as: cartridges, columns or membranes containing an extractive absorbent.
The sample is inserted into the cartridge and all the matrix interferences that have been adsorbed are washed by a weak solvent; the elution of the analytes is done using a small amount of the appropriate solvent [1]. The crucial factors to apply these analytical techniques are the choice of the solvent linked to the affinity for analytes, the toxicity, the selectivity, the volatility and finally the price, that must not be underestimated.
The search for rapid, efficient, cost effective and environment-friendly means of analytical extractions has prompted the introduction, in 1989 of Solid Phase Microextraction (SPME) by J. Pawliszyn [2]. Compared to traditional analytical techniques, SPME is faster, simpler, more sensitive and does not require an extraction solvent. Since the invention of the technique until today, its applications have dramatically increased. Therefore to date its application are numerous, and include the determination of natural products, also in trace, in drinking, fruits and biological samples [2].
Finally dispersive solid phase extraction (d-SPE) analysis is a simple and straightforward sample preparation technique suitable for a wide variety of food and agricultural products. One specific method which has become popular is the QuEChERS technique which was created to facilitate the rapid screening of large numbers of food and agricultural samples for pesticide residues.
It could be a valid alternative to SPME, but according to our knowledge it is not among the methods of choice for the determination of fat-soluble vitamins
In the light of these considerations in this paper the SPME has been chosen for the scope of the experimental work described in this paper that aims to develop a rapid, sensitive and specific analytical method for the simultaneous determination of fat-soluble vitamins E and K in food and biological matrices.
2.Comment of reviewer: Section 2.2 does not present results. It should be moved the Materials and Methods section.
2.Answer: We moved the section in Material and Methods
3.Comment of reviewer:The chromatograms presented in Figure 4A and 4B show a baseline anomaly in the region around peaks 1 and 2. Any idea of the cause of this?
3.Answer: To compare resolution in problematic segment in Figure 4A e 4B, a zoom (x 10) is plotted. This variation was described in the caption
Reviewer 2 Report
The article from Corbo et al. describes three different treatments for the simultaneous determination of Vitamin E and Vitamin K. The extraction procedures are optimized, and the resulting analytical methods completely validated. From the scientific point of view, this reviewer has only one comment as the research has been seriously developed.
- Table 3 must be better explained. What is the type of recovery (absolute or relative) presented in the Table? That is a critical factor. This reviewer guesses that the recovery presented is the absolute value (percentage of analyte that is effectively isolated from the sample). If this is correct, please just write absolute before recovery.
Other minor comments are stated below
- Line 50. Reference (SPME) missing.
- Line 52. The stamen “requiring only water for the extraction” needs further explanation.
- Use the acronyms once they have been defined. There are plenty of examples for that Lines 75, 84, 85, 95..
- Line 100. Define the acronym (LOD).
- The quality of the Figures must be improved. In some of them, shadows and lines in the left are visible.
Author Response
The article from Corbo et al. describes three different treatments for the simultaneous determination of Vitamin E and Vitamin K. The extraction procedures are optimized, and the resulting analytical methods completely validated. From the scientific point of view, this reviewer has only one comment as the research has been seriously developed.
We thank the Reviewer for the constructive comments on our manuscript
1. Reviewer Comment: Table 3 must be better explained. What is the type of recovery (absolute or relative) presented in the Table? That is a critical factor. This reviewer guesses that the recovery presented is the absolute value (percentage of analyte that is effectively isolated from the sample). If this is correct, please just write absolute before recovery.
1 Answer: Those indicated are the tested concentrations, the values shown are% recoveries. For example we have subjected to treatment A a sample containing the vit. at a concentration of 5 ug / ml, 59 +/- 16% was recovered We wrote % before each sample.
2.Reviewer Comment: Other minor comments are stated below
2 Answer: all revisions are described point by point in the following table
- Line 50. Reference (SPME) missing. The reference 2 is inserted
- Line 52. The stamen “requiring only water for the extraction” needs further explanation. We inserted the explanation in the introduction that was improved
- Use the acronyms once they have been defined. There are plenty of examples for that Lines 75, 84, 85, 95. Done
- Line 100. Define the acronym (LOD). The sentence Limit of detection is inserted
- The quality of the Figures must be improved. In some of them, shadows and lines in the left are visible. The quality of Figures is improved
|
Line |
Pag. |
Before change |
After change |
|
60 |
2 |
The reference missing inserted |
[2] |
|
106 |
3 |
Solid Phase Microextraction |
based SPME |
|
115 |
3 |
dispersive liquid–liquid microextraction |
DLLME |
|
131 |
3 |
- |
limit of detection |
Reviewer 3 Report
This paper describes the development of a method that is intended to enable the merging of different approaches for the analysis of Vitamins in vegetable produce. The advantages of SPME (high-throughput and low solvent consumption) are well know in academic research, but the technology is still not used as widely as it should in industry and government regulatory agencies because of the perception that quantitative analysis is not as robust as traditional sample preparation techniques such as SPE and LLE. As such, it is a worthwhile effort to demonstrate the equivalence of the non-exhaustive SPME extractions with the exhaustive extractions of the conventional techniques. This was adequately demonstrated in this project for a decent variety of vegetable samples (carrots, onions, celery, kale), and therefore I support the publication of this manuscript.
There are some changes to the manuscript that I would like to recommend to the authors:
- With regards to Figure 4, I would like to see the NIST library match values for 4 target compounds rather than the mass spectra that are shown. The library match provides a better confidence metric of the similarity of the spectra. In addition, no information if provided for the 4th compound.
- A comment should be made to specifically address the throughput and low-solvent consumption advantages of the SPME method (perhaps somewhere in the discussion section). How much faster is the SPME protocol than the others. How much solvent is used by SLE-LLE, for example, and how does this accumulate over a period of time or a significant number of samples?
- The quality of the written material in the manuscript can be improved in a number of sentences. I will provide some examples that I happened to catch:
- Line 31 (in the abstract): “which makes it more close to…” should be replaced with “which makes it the preferable option for…”
- Line 52: “numerous, also for the determination...” should be replaced with “numerous, and include the determination…”
- Line 57: “Human” should be “Humans”
- Line 60: “by EFSA” should be “by the EFSA”
- Line 61: “value, like for example, extra virgin…” can be replaced with “value, e.g. extra virgin…”
- Line 69: “and its lipophilicity. These challenges…” should be replaced with “and its lipophilicity, but these challenges…”
- Line 71: “Existing methods for its determination in food are mainly…” should be “Existing methods for the determination of vitamins in food are mainly…”
- Line 76: “solid phase microextraction (SPME)” should be replaced with “SPME”. SPME has already been abbreviated in line 50, so it should only be referred to from that point by its acronym.
- Line 84: “dispersive liquid liquid microextraction” should be replaced with “DLLME”, which has already been abbreviated in line 79.
- Line 107 (Results section): “in CHRIST-vacuum concentrator” should be “in a CHRIST-vacuum concentrator”.
- Line 140: “The method was firstly developed working…” should be “the method was initially developed working…”
- Line 144: “Then, the chromatographic conditions were…” should be “The chromatographic conditions were then…”
- Lines 145-146: “was obtained directly injecting a standard solution…” should be “was obtained by directly injecting a standard solution…”
- Lines 149-150: “greater than 0.999 and intercept not significantly different from zero at 95% confidence level.” should be “greater than 0.999 and intercepts significantly close to zero at the 95% confidence level.”
In summary, this manuscript is a good proof-of-concept report of the combination of different methods with a high-throughput and “green” sample preparation approach. It would be nice to comment (perhaps in the conclusion” that this approach could be extended to more target compounds and more Vitamins (Vitamin A for instance) in future work.
Author Response
Reviewer's comments:This paper describes the development of a method that is intended to enable the merging of different approaches for the analysis of Vitamins in vegetable produce. The advantages of SPME (high-throughput and low solvent consumption) are well know in academic research, but the technology is still not used as widely as it should in industry and government regulatory agencies because of the perception that quantitative analysis is not as robust as traditional sample preparation techniques such as SPE and LLE. As such, it is a worthwhile effort to demonstrate the equivalence of the non-exhaustive SPME extractions with the exhaustive extractions of the conventional techniques. This was adequately demonstrated in this project for a decent variety of vegetable samples (carrots, onions, celery, kale), and therefore I support the publication of this manuscript.
There are some changes to the manuscript that I would like to recommend to the authors:
1.Reviewer's comments: With regards to Figure 4, I would like to see the NIST library match values for 4 target compounds rather than the mass spectra that are shown. The library match provides a better confidence metric of the similarity of the spectra. In addition, no information if provided for the 4thcompound.
1.Answer: The NIST library values for α-tocopherol, α-tocopheryl acetate and phylloquinone were 830, 850 and 813, respectively, and were added in the caption of Figure 4. Compound IV is vitamin K2 or menaquinone, which is used in this study as an internal standard, because it is usually found in animal or fermented products, as specified in text
2.Reviewer's comments A comment should be made to specifically address the throughput and low-solvent consumption advantages of the SPME method (perhaps somewhere in the discussion section). How much faster is the SPME protocol than the others. How much solvent is used by SLE-LLE, for example, and how does this accumulate over a period of time or a significant number of samples?
2.Answer: According the suggestion of reviewer we inserted this sentence in conclusion section
- In conclusion, although the benefits of SPME (high productivity and low solvent consumption) are well known in academic research, the technology is not yet used as widely as it should by regulatory agencies, perhaps due to the perception that quantitative analysis is not solid. as much as traditional techniques. In this study, however, the equivalence of the non-exhaustive SPME with the exhaustive extractions of the conventional SLE and LLE techniques is demonstrated. Sample preparation is much simpler (0.075 g of sample in 15 ml of water with 10% ethanol) and faster (extraction time with the fiber of only 30 minutes, fully automatic) compared to the other two conventional procedures.
- 3.Reviewer's comments The quality of the written material in the manuscript can be improved in a number of sentences. I will provide some examples that I happened to catch:
- Line 31 (in the abstract): “which makes it more close to…” should be replaced with “which makes it the preferable option for…”
- Line 52: “numerous, also for the determination...” should be replaced with “numerous, and include the determination…”
- Line 57: “Human” should be “Humans”
- Line 60: “by EFSA” should be “by the EFSA”
- Line 61: “value, like for example, extra virgin…” can be replaced with “value, e.g. extra virgin…”
- Line 69: “and its lipophilicity. These challenges…” should be replaced with “and its lipophilicity, but these challenges…”
- Line 71: “Existing methods for its determination in food are mainly…” should be “Existing methods for the determination of vitamins in food are mainly…”
- Line 76: “solid phase microextraction (SPME)” should be replaced with “SPME”. SPME has already been abbreviated in line 50, so it should only be referred to from that point by its acronym.
- Line 84: “dispersive liquid liquid microextraction” should be replaced with “DLLME”, which has already been abbreviated in line 79.
- Line 107 (Results section): “in CHRIST-vacuum concentrator” should be “in a CHRIST-vacuum concentrator”.
- Line 140: “The method was firstly developed working…” should be “the method was initially developed working…”
- Line 144: “Then, the chromatographic conditions were…” should be “The chromatographic conditions were then…”
- Lines 145-146: “was obtained directly injecting a standard solution…” should be “was obtained by directly injecting a standard solution…”
- Lines 149-150: “greater than 0.999 and intercept not significantly different from zero at 95% confidence level.” should be “greater than 0.999 and intercepts significantly close to zero at the 95% confidence level.”
3.Answer: all revisions are reported point by point in the following table
|
Line |
Pag. |
Before change |
After change |
|
32 |
1 |
which makes it more close to |
which makes it the preferable option for |
|
63 |
2 |
numerous, also for the determination |
numerous, and include the determination |
|
79 |
2 |
Human |
Humans |
|
82 |
2 |
EFSA |
the EFSA |
|
83 |
2 |
value, like for example, extra virgin |
value, e.g. extra virgin |
|
91 |
2 |
and its lipophilicity. These challenges |
and its lipophilicity, but these challenges |
|
93 |
2 |
Existing methods for its determination in food are mainly |
Existing methods for the determination of vitamins in food are mainly |
|
106 |
3 |
Solid Phase Microextraction |
based SPME |
|
115 |
3 |
dispersive liquid–liquid microextraction |
DLLME |
|
140 |
3 |
- |
a |
|
143 |
4 |
The method was firstly developed working |
The method was initially developed working |
|
149 |
4 |
Then, the chromatographic conditions were |
The chromatographic conditions were then |
|
150 |
4 |
was obtained directly injecting a standard solution |
was obtained by directly injecting a standard |
|
155 |
4 |
greater than 0.999 and intercept not significantly different from zero at 95% confidence level |
greater than 0.999 and intercepts significantly close to zero at the 95% confidence level. |
4.Reviewer's Comment:
In summary, this manuscript is a good proof-of-concept report of the combination of different methods with a high-throughput and “green” sample preparation approach. It would be nice to comment (perhaps in the conclusion” that this approach could be extended to more target compounds and more Vitamins (Vitamin A for instance) in future work
4.Answer
We appreciated your suggestion and therefore we inserted a sentence in the conclusion:
In summary, in this manuscript a good proof-of-concept report of the combination of different methods with a high-throughput and “green” sample preparation approach is reported. This approach could be extended to more target compounds and more Vitamins (Vitamin A for instance) in future work.